# Weathering the storm of COVID-19 pandemic: A cross-sectional survey of reported changes in first contact physiotherapy services in the UK and Australia

**Oluwatoyin Adenike Adeniji**[1,2]*, **Evangelos Pappas**[3,4], **Karen Stenner**[1], **Victoria Traynor**[5], **Nicola Carey**[6], **Theopisti Chrysanthaki**[1]

**1** School of Health Sciences, Faculty of Health and Medical Sciences, University of Surrey, Guildford, United Kingdom, **2** School of Medical, Indigenous and Health Sciences, Faculty of Science Medicine and Health, University of Wollongong, Wollongong, Australia, **3** School of Health and Biomedical Sciences, Royal Melbourne Institute of Technology University, Melbourne, Australia, **4** Sydney School of Health Sciences, The University of Sydney, Camperdown, Australia, **5** School of Health, University of the Sunshine Coast, Sippy Downs, Australia, **6** Centre for Rural Health Sciences, University of the Highlands and Islands, Inverness, United Kingdom

* o.adeniji@surrey.ac.uk

## Abstract

Limited evidence exists on early COVID-19 related changes in First Contact Physiotherapy Services (FCPS) for musculoskeletal (MSK) patients within the UK primary care (PC) and Australian emergency departments (ED), knowledge that is crucial for understanding their level of responsiveness and readiness for future crises. This study explores the initial changes in FCPS during the COVID-19 pandemic in the UK and Australia. The UK and Australia represent a function of both country and their FCPS context (UK [PC], Australia [ED]). A cross-sectional survey was conducted from January-April 2023. Data were self-reported by physiotherapists in FCPS roles, managing MSK patients in the UK and Australia. Only responses from those who recalled changes in FCPS were included, with 153 participants analysed. Descriptive statistics and two-way ANOVA were used to examine the effects of timing of change, country, and their interaction on readiness and responsiveness to MSK patient needs. Overall, 75.7% of initial changes were perceived to have occurred within three months following the World Health Organisation's declaration of COVID-19 as a global pandemic. Participants from both countries differed significantly in their perceptions of how COVID-19 affected patient access to FCPS (p < 0.001). Changes in MSK patient presentation to FCPS varied significantly by both the timing of the change and the country (p < 0.001). Similarly, changes in the care delivery platform were significantly associated with the timing of change (p = 0.014) and the country (p < 0.001). While participants' responses generally indicated inadequate readiness, the overall perceived responsiveness was higher in the UK (using an arbitrary cut-off of ≥50%) compared to Australia. No significant effects of timing or country were found

**Data availability statement:** All relevant, non-identifiable data are contained within the manuscript. Raw data could be accessed through the University of Surrey, UK open access repository. https://doi.org/10.15126/surreydata.901476.

**Funding:** This study is part of a PhD project, jointly funded by the University of Surrey Doctoral College Studentship and the University of Wollongong Higher Degree Research Studentship, through a collaboration called University Global Partnership Network. In addition, the publication fee for this article was covered by the University of Surrey.

**Competing interests:** The authors have declared that no competing interests exist.

on either readiness or responsiveness. This highlights the need to strengthen FCPS readiness for more responsive future crises. It also suggests the need for contextual considerations when developing readiness strategies.

## Introduction

COVID-19, declared a global pandemic on March 11, 2020 by the World Health Organisation (WHO), caused major disruptions to the delivery of healthcare services across the world [1,2]. Primary care (PC) and emergency department (ED) services were particularly impacted as resources were directed towards caring for those with COVID-19 virus infection [1,3–7], impacting other essential non-COVID-19 care services such as first contact physiotherapy services (FCPS) within these settings [8,9]. FCPS provide the first point of contact for the assessment of musculoskeletal (MSK) patients, including immediate management, ongoing referral, or advice. FCPS play a crucial role in the care of MSK patients [10,11], accounting for about 20% of the UK PC [12] and 30% of Australia ED presentations [13].

The UK and Australia are two of the leading countries providing FCPS in PC and ED settings, respectively [10,11,14]. Each country responded differently to the COVID-19 pandemic: in the UK, the initial national response during the first wave, between March 16th and 23rd, 2020, included a lockdown, with the public advised to stay at home to limit the spread of infection and alleviate pressure on healthcare resources [15,16]. Significant changes were reported across all healthcare services, including PC, where general practitioners (GPs) were instructed to prioritise urgent and serious conditions, suspend routine appointments, and supplement care with the use of telehealth services [6,17,18], with impacts on staff and patients [19–24].

In Australia, the pandemic response was swift, with the government announcing the Australian health sector emergency response plan on February 27, 2020, before COVID-19's declaration as a global pandemic. Measures included monitoring and minimising community transmission [25]. A series of lockdown measures were implemented across Australia states with varied timing, including the closure of non-essential businesses and activities [26,27]. Due to increased number of COVID-19 cases, many non-urgent health services were suspended, and telehealth services including government funded were introduced to minimise human contact [28,29]. EDs, where COVID-19 infected patients were mainly managed, experienced disruption due to strategic changes including service redesign [30–32].

While studies have shown disruption in PC and ED services followed by the introduction of public health intervention policies in early 2020 [1,4,6,7,18,31,33], minimal consideration has been given to specific services such as FCPS provided within these settings. Evidence from the USA and Italy highlighted reduced access and presentation of MSK patients in ED [8,9]. Additionally, the use of telehealth for MSK services in UK PC where decisions were made without planning or guidance may have led to challenges in effective implementation [34]. The lack of preparation may have also impacted responsiveness, relating to meeting individuals' healthcare

expectations [35,36]. Understanding these changes offers valuable lessons to support FCPS' agility in responding to unforeseen challenges, strengthening preparedness strategies, and preventing or minimising disruption. Additionally, understanding the timeline of these changes is crucial for assessing the readiness and responsiveness of FCPS during the COVID-19 pandemic. This insight could inform preparedness strategies for future crises [37]. Therefore, this study aims to explore changes in FCPS during the COVID-19 pandemic in the UK and Australia, with a focus on the initial changes. This approach helps to gauge the speed of FCPS response and readiness based on physiotherapists' self-reported experiences in the FCPS role. For this study, "changes" refers to the transformations or modifications in the delivery of healthcare by FCPS to patients with MSK conditions during the COVID-19 pandemic [38].

### Study aim

This study aims to explore the initial changes in FCPS during the COVID-19 pandemic in the UK PC and Australian ED.

## Methods

### Study design

A cross-sectional study was conducted using an online survey instrument hosted by Qualtrics and distributed to physiotherapists managing MSK patients at first contact in PC and ED in the UK and Australia, respectively. The survey ran from 16th January to 30th April 2023. Reporting follows the Checklist for Reporting Results of Internet E-Surveys (CHERRIES) [39]. Completed checklist can be found in Appendix S1 Checklist.

### Ethical approvals

Favourable ethical opinion has been obtained from the research governance office, at the University of Surrey UK (FHMS 21–22 136 EGA) and further approval was received from the Human Research Ethics committee at the University of Wollongong (UOW), Australia (UOW22/163). Electronic informed consent which was embedded in the survey was obtained from participants before proceeding with the survey.

### Sample and recruitment

A non-probability sampling method was employed [40], using contacts with physiotherapy professional organisations and accessing social media to recruit 211 physiotherapists managing MSK patients at first contact in the UK and Australia. The eligibility criteria included physiotherapists who had been practicing in an FCPS role for MSK patients in the UK PC or Australian ED before or since March 2020. Participants also needed to be aware of and able to recall changes that occurred in FCPS during the COVID-19 pandemic. The online survey link was advertised and distributed to physiotherapists working within FCPS through various platforms, including interactiveCSP | The Chartered Society of Physiotherapy (an online network for physiotherapists, including those practising in PC for MSK in the UK), the Australia Physiotherapy Association newsletters as well as Facebook groups, Twitter (now X) and LinkedIn.

### Survey instrument

In the absence of an instrument designed to assess changes in health services, including FCPS, particularly during public health emergencies such as COVID-19, the primary researcher (OA) developed a survey to identify initial changes in FCPS including the timeline, as perceived by physiotherapist in FCPS role. The survey was developed with support from the team (Appendix S2 Survey) [37]. The survey comprises of 53-items, divided into three sections, using a mix of implementation science and responsiveness frameworks and findings from the literature on readiness for changes in healthcare. The initial page of the survey introduced the study to participants by providing detailed information about the study through an embedded participant's information sheet and a consent form. The consent was set up as a requirement

for participants to proceed with the survey. Section one was developed to collect data on the changes in FCPS, informed by the Framework for Reporting Adaptation and Modification Enhancement (FRAME) [41]. This section consists of 33 questions with conditional display and skips functions, including an eligibility question about the participant's awareness of changes in their FCPS during the COVID-19 pandemic, which determines whether the participant continues with the survey or exits. The second section contains two questions on readiness and responsiveness, developed based on findings from the literature on readiness [42] and the WHO's responsiveness framework, which was adapted to also include equitability [35]. These questions aim to understand FCPS readiness and responsiveness to change during the COVID-19 pandemic from the practitioner's perspective. The third section gathers data on general demographics and employment information. The survey was designed in such a way that only questions requiring conditional display were mandatory for participants to answer.

Two versions of the questionnaire were created with similar contents but catering for the contextual differences in the delivery of FCPS in UK and Australia (Appendix S2 Survey) [37]. To assess whether the sequence, flow and content of survey questions were comprehensible, coherent and appropriate and to identify any problems before main data collection, the survey instrument was pilot tested with 12 physiotherapists in FCPS role in the UK (n=7) and Australia (n=5). The 12 participants hold master's degrees, with 75% (n=9) having more than four years of experience in FCPS, and 25% (n=3) having three years of experience as at the time of data collection. The pilot survey was disseminated using a Qualtrics link and employed a variety of response style questions (e.g., Likert scales, dichotomous questions), including open comments for feedback on the content, format and structure of the questionnaire. Eight responses were received in total. In addition, comprehension and time required to complete the survey were assessed; the completion time averaged 13.9 minutes (SD 10.0), with a 100% completion rate. Participants found the questionnaire content relevant to FCPS and the research question. Some terminology adjustments and clarification of the survey's time frame were suggested. To address time frame concerns, two participants were consulted by telephone, highlighting inconsistencies in relation to phrases like "when the change first occurred," "during COVID," and "due to COVID." These issues were subsequently resolved, and adjustments were made to enhance question clarity and structure.

**Reliability of scale**

A series of reliability analyses were carried out using Cronbach alpha coefficients to determine the internal consistency and reliability of the scales used to measure the type of change, readiness, and responsiveness (Table 1).

**Data analysis**

The analysis of survey responses by country reflects a function of both country and their FCPS context (UK [PC] and Australia [ED]). Data from Qualtrics for both the UK and Australia was downloaded into Excel, merged, cleaned, and keyed into the data editor of SPSS version 28© (IBM). First, a descriptive analysis of frequencies, and percentages was

**Table 1. Reliability of scale using Cronbach alpha coefficients.**

| Domains | M | SD | Cronbach's alpha | No of items |
|---|---|---|---|---|
| Changes in access | 22.9 | 4.19 | 0.68 | 6 |
| Changes in platform for delivery | 6.53 | 2.79 | 0.74 | 3 |
| Changes in MSK conditions managed | 13.9 | 3.03 | 0.81 | 6 |
| Changes in interventions provided | 20.9 | 3.67 | 0.60 | 9 |
| Changes in caseload | 4.03 | 1.15 | 0.74 | 3 |
| Readiness for change | 46.1 | 11.0 | 0.93 | 15 |
| Responsiveness with change | 38.1 | 6.57 | 0.84 | 11 |

MSK: Musculoskeletal.

used to summarise participant characteristics, timing and type of change, readiness and responsiveness by country. The timing of change was categorised into two periods: within three months of the COVID-19 pandemic declaration (March to June 2020) and over four months (July 2020 onwards). The categorisation into two periods was informed by a systematic integrative review which shows a similar trend [36]. To determine whether the timing of change influenced the type of changes, levels of readiness and responsiveness, a two-way ANOVA was carried out. The significance level was set a priori at 0.05. A composite score of mean was calculated using a scale of 1 to −1 (increased 1, same 0, decreased −1) for changes and (agreement 1, neutral 0, disagreement −1) for readiness and responsiveness. An arbitrary cut-off of 50% agreement was used to determine the level of readiness and responsiveness (≥50% indicating moderate to high agreement, <50% indicating low agreement). While timing and severity of COVID-19 waves and responses varied across countries, regions, and even states, the decision to use the WHO's declaration of COVID-19 as a global pandemic (March 11, 2020) as a reference point was based on its symbolic and practical value as a widely recognised turning point in public and institutional awareness, often triggering more formalised actions, including lockdowns, and shifts in healthcare delivery. Although the implementation of these measures varied, insights from our preliminary one-to-one consultations during the study's conceptualisation phase, along with existing literature, indicated that the WHO's declaration served as a common temporal anchor [43] in both professional and public consciousness. Using this reference point also help us enhance comparability across diverse settings, particularly in an international study where more context-specific time frames may complicate analysis and interpretation.

## Results

The results presented by country represent a function of both country and FCPS context (UK [PC] and Australia [ED]). Findings are reported in line with SAMPL Guidelines [44].

### Participant characteristics

Two hundred and eleven individuals accessed the electronic consent form, with 94% (198/211) providing their consent. Among those who consented, 96% (192/198) progressed to the eligibility question, inquiring about their awareness of changes in FCPS. Of this group, 67.7% (130/192) indicated awareness of the changes, while 12% (23/192) were uncertain, and 20.3% (39/192) were unaware of FCPS changes. Most exclusions due to lack of awareness of changes occurred among Australian participants (82.1% or 32/39). In total, 76.7% (153/192) were included in the analysis, comprising 64.1% (98/153) from the UK and 35.9% (55/153) from Australia. A summary of participant characteristics can be found in Table 2.

### Changes in FCPS

**Types of changes.** Suspension of FCPS was reported by 40.6% (n = 58) of study participants, with the majority (71.2%, n = 42) occurring between March and June 2020, the first three months following the declaration of COVID-19 as a global pandemic. Among UK respondents, 60.0% (n = 53) reported suspensions, compared to only 9.3% (n = 5) of respondents from Australia. In general, physiotherapists perceived that the initial changes in the delivery of FCPS occurred within the first three months 75.7% (n = 109), with 24.3% (n = 44) occurring after four months or longer across both countries. Perceived changes in FCPS varied by country; among the UK participants, 59.2% (n = 42) observed an increased use of triaging process compared to 14.3% (n = 5) observed by Australian respondents. Respondents also perceived an increased use of telephone services for patients requesting appointments in the UK 77.9% (n = 67) compared to Australia 4.3% (n = 1). Furthermore, increased use of remote (telehealth only) 87.8% (n = 65) and hybrid care (in-person and remote) 56.8% (n = 42) was reported in the UK, when compared to Australia 20.5% (n = 9), 21.7% (n = 10), respectively. FCPS in Australia was predominantly reported to remain face-to-face 45.5% (n = 20); however, a decreased in face-to-face care was observed in both countries (UK 78.4% (n = 58), Australia 54.5% (n = 24)). In terms of intervention

**Table 2. Participant characteristics.**

| | UK n (%) | Australia n (%) | Total n (%) |
|---|---|---|---|
| **Gender** | | | |
| Female | 44 (64.7) | 52 (51.0) | 64 (59.3) |
| Male | 24 (35.3) | 48 (49.0) | 44 (40.7) |
| **Age group** | | | |
| 18-24 | 0 (0.00) | 0 (0.00) | 0 (0.00) |
| 25-34 | 12 (17.6) | 6 (14.6) | 18 (16.5) |
| 35-44 | 30 (44.1) | 16 (39.0) | 46 (42.2) |
| 45−5 | 20 (29.4) | 14 (34.1) | 34 (31.2) |
| 55-64 | 6 (8.80) | 5 (12.2) | 11 (10.1) |
| **Academic qualification** | | | |
| Diploma | 2 (2.90) | 2 (4.90) | 4 (3.70) |
| Bachelor | 18 (26.5) | 14 (14.6) | 24 (22.0) |
| Masters | 46 (67.6) | 31 (75.6) | 77 (70.7) |
| Doctorate | 2 (2.90) | 2 (4.90) | 4 (3.70) |
| **Experience as physiotherapists in FCPS role** | | | |
| ≥ 10 years | 0 (0.00) | 10 (24.2) | 10 (9.20) |
| 5–9 years | 30 (44.1) | 17 (41.5) | 47 (43.1) |
| 3–4 years | 38 (55.9) | 14 (34.2) | 52 (47.7) |
| **% of time spent each week as a physiotherapist in FCPS role treating MSK** | | | |
| 0-25% | 8 (11.8) | 4 (9.80) | 12 (11.0) |
| 26-50% | 16 (23.5) | 6 (14.6) | 22 (20.2) |
| 51-75% | 10 (14.7) | 16 (39.0) | 26 (23.9) |
| 76-100% | 34 (50.0) | 15 (36.6) | 49 (45.0) |
| **Region of practice in the UK** | | | |
| England | 54 (78.3) | – | – |
| Scotland | 9 (13.0) | – | – |
| Wales | 4 (5.80) | – | – |
| Northern Ireland | 2 (2.90) | – | – |
| **Region of practice in Australia** | | | |
| New South Wales | – | 18 (41.9) | – |
| Victoria | – | 12 (27.9) | – |
| Western Australia | – | 9 (20.9) | – |
| Queensland | – | 4 (9.30) | – |

FCP: First contact physiotherapist, MSK: Musculoskeletal conditions.

n=153 (UK = 98 (64.1%), Australia = 55 (35.9%)).

Missing data account for the discrepancies in frequency n, valid percentage reported in this Table.

provided during the pandemic, UK respondents reported increased use of self-management 86.8% (n = 59), compared to Australia 44.7% (n = 21). Details can be found in Table 3.

**Readiness and responsiveness with changes:** In general, strong agreement was consistently reported for the readiness factors of access to information and leadership, while stakeholder's buy-in recorded no agreement across both the UK and Australia. In the UK, participants reported moderate to high agreement (≥50%) with the following four readiness items, presented in descending order of agreement: access to information (72.7%, n = 48), leadership (65.5%,

**Table 3. Descriptive Statistics of Physiotherapists' Perceptions of Initial Changes in FCPS.**

| Changes (Domains) Items | UK | | | | | Australia | | | | |
|---|---|---|---|---|---|---|---|---|---|---|
| | Increased n(%) | Same n(%) | Decreased n(%) | Never used n(%) | n (%) | Increased n(%) | Same n(%) | Decreased n(%) | Never used n(%) | n (%) |
| **Access** | | | | | | | | | | |
| Triage | 42(59.2) | 21(29.6) | 1(1.40) | 7(9.90) | 71(67.2) | 5(14.3) | 23(65.7) | 7(20.0) | 0(0.00) | 35(33.0) |
| In-person/self-referral/Walkin | 2(2.90) | 21(30.0) | 21(30.0) | 26(37.1) | 70(67.3) | 2(5.9) | 9(26.5) | 18(52.9) | 26(37.1) | 34(32.7) |
| Online self - booking | 7(10.6) | 11(16.7) | 5(7.60) | 43(65.2) | 66(66.0) | 0(0.00) | 2(5.90) | 1(2.90) | 31(91.0) | 34(34.0) |
| Ambulance service | 2(3.20) | 11(17.7) | 2(3.20) | 47(75.8) | 62(65.3) | 2(6.10) | 15(45.5) | 20(30.3) | 6(18.2) | 33(34.7) |
| FCP self-select | 13(19.1) | 11(16.2) | 3(4.40) | 41(60.3) | 68(68.0) | 1(3.10) | 16(18.6) | 10(31.1) | 0(0.00) | 32(32.0) |
| Telephone | 67(77.9) | 16(18.6) | 3(3.50) | 0(0.00) | 86(78.9) | 1(4.30) | 3(13.0) | 1(4.30) | 18(78.3) | 23(21.1) |
| **Platform for delivery** | | | | | | | | | | |
| Remotely via telehealth | 65(87.8) | 3(4.10) | 2(2.70) | 4(5.40) | 74(62.7) | 9(20.5) | 3(6.8) | 2(4.50) | 30(68.2) | 44(37.3) |
| Face to face/in-person | 6(8.10) | 7(9.5) | 58(78.4) | 3(4.10) | 74(62.7) | 0(0.00) | 20(45.5) | 24(54.6) | 0(0.00) | 44(37.3) |
| Hybrid | 42(56.8) | 11(14.9) | 14(18.9) | 7(9.50) | 74(61.7) | 10(21.7) | 3(6.50) | 2(4.30) | 31(67.4) | 46(38.3) |
| **MSK conditions presentation** | | | | | | | | | | |
| Soft tissue injury | 6(9.50) | 36(57.1) | 21(33.3) | 0(0.00) | 63(56.3) | 1(2.00) | 6(12.2) | 42(85.7) | 0(0.00) | 49(43.8) |
| Arthritis | 15(23.8) | 40(47.6) | 18(28.6) | 0(0.00) | 63(56.3) | 0(0.00) | 9(18.4) | 40(81.6) | 0(0.00) | 49(43.8) |
| Spine related pain | 19(30.2) | 28(44.4) | 16(25.4) | 0(0.00) | 63(56.3) | 0(0.00) | 20(40.8) | 29(44.1) | 0(0.00) | 49(43.8) |
| Dislocation/minor fracture | 2(3.40) | 20(40.8) | 26(44.1) | 11(18.6) | 59(55.1) | 2(4.20) | 2(18.8) | 35(72.9) | 2(18.8) | 49(44.9) |
| Post orthopaedic surgical conditions | 6(9.50) | 4(6.30) | 36(57.1) | 17(27.0) | 63(56.3) | 0(0.00) | 2(4.10) | 41(83.7) | 6(12.2) | 49(43.8) |
| Muscles & ligament pain | 7(11.1) | 36(57.1) | 20(31.7) | 0(0.00) | 63(56.3) | 1(2.00) | 6(12.2) | 42(85.7) | 0(0.00) | 49(43.8) |
| **Interventions provided** | | | | | | | | | | |
| Self-management | 59(86.8) | 9(13.2) | 0(0.00) | 0(0.00) | 68(59.1) | 21(44.7) | 23(48.9) | 3(6.40) | 0(0.00) | 47(40.9) |
| Referral to physio | 12(18.5) | 8(12.3) | 45(69.2) | 0(0.00) | 65(58.6) | 0(0.00) | 20(43.5) | 26(56.5) | 0(0.00) | 46(41.4) |
| Referral to comm services | 12(18.5) | 16(24.6) | 35(53.8) | 2(3.10) | 65(58.6) | 2(4.30) | 15(32.6) | 29(63.0) | 0.(0.00) | 46(41.4) |
| Referral to other specialist | 11(16.9) | 26(40.0) | 28(43.1) | 0(0.00) | 65(60.2) | 2(4.70) | 20(46.5) | 21(48.8) | 0(0.00) | 43(39.8) |
| Application of orthosis | 0(0.00) | 18(30.5) | 23(39.0) | 18(30.5) | 59(56.2) | 0(0.00) | 35(76.1) | 11(23.9) | 0(0.00) | 46(43.8) |
| Surgery booking | 0(0.00) | 10(18.1) | 27(50.0) | 17(31.5) | 54(54.0) | 4(8.70) | 20(28.5) | 25(54.3) | 17(31.5) | 46(46.0) |
| Prescribing imaging | 10(15.4) | 25(38.8) | 22(33.8) | 25(38.5) | 65(58.6) | 8(17.4) | 34(73.9) | 8(17.4) | 0(0.00) | 46)41.1) |
| Prescribing medication | 9(15.5) | 26(44.8) | 3(5.20) | 20(34.5) | 58(100) | – | – | – | – | – |
| Prescribing/administering injection | 2(3.20) | 6(9.50) | 41(65.1) | 14(22.2) | 63(100) | – | – | – | – | – |
| **Caseload** | | | | | | | | | | |
| Ave no of MSK/week | 14(21.3) | 14(21.2) | 38(57.6) | – | 66(59.5) | 2(4.40) | 0(0.00) | 43(95.6) | – | 45(40.5) |
| Ave no of consult time/week | 29(46.0) | 26(41.3) | 8(12.7) | – | 63(57.3) | 26(55.3) | 15(31.9) | 6(12.8) | – | 47(42.7) |

*(Continued)*

**Table 3.** (Continued)

| Changes (Domains) Items | UK | | | | | Australia | | | | |
|---|---|---|---|---|---|---|---|---|---|---|
| | *Increased n(%)* | *Same n(%)* | *Decreased n(%)* | *Never used n(%)* | *n (%)* | *Increased n(%)* | *Same n(%)* | *Decreased n(%)* | *Never used n(%)* | *n (%)* |
| Ave no of app/patient | 27(42.9) | 15(23.8) | 21(33.3) | – | 64(63.6) | 0(0.00) | 22(61.1) | 14(38.9) | – | 36(36.4) |
| Administrative/managerial duties | 32(48.5) | 30(45.5) | 4(6.10) | – | 63(63.5) | 26(68.4) | 11(28.9) | 1(2.60) | – | 36(36.4) |

FCPS: First contact physiotherapy services, MSK: Musculoskeletal, Ave no: Average number, app: appointment.

This Table presents observed initial changes in FCPS in terms of the level of use of different FCPS access route, platforms for delivering care, interventions provided, including the levels of patient presentation, and staff workload, as perceived by physiotherapists working in FCPS roles in UK primary care and Australian emergency departments.

n = 42), mission and vision (55.2%, n = 32), and output and results (51.7%, n = 30). The remaining items showed lower levels of agreement (<50%), including: compatibility (48.5%, n = 32), infrastructure (39.4%, n = 26), external partnerships and support (39.3%, n = 22), financial resources (38.1%, n = 16), learning strategy (35.7%, n = 20), supplies and procurement (35.5%, n = 32), availability of emergency response guide (35.5%, n = 22), human resources (37.9%, n = 22), human resources management (20.7%, n = 12), technical support (6.1%, n = 4), and stakeholders' buy-in (0%). In Australia, moderate to high agreement (≥50%) was also reported for three readiness items, listed in descending order: supplies and procurement (73.7%, n = 28), access to information (63.4%, n = 26), and leadership (53.7%, n = 22). All other items recorded lower levels of agreement (<50%), including availability of emergency response guide (48.7%, n = 19), financial resources (43.2%, n = 16), mission and vision (41.0%, n = 16), human resources management (46.2%, n = 18), learning strategy (40.5%, n = 15), human resources (39.0%, n = 16), compatibility (32.4%, n = 12), external partnership and support (24.2%, n = 8), output and results (19.5%, n = 8), infrastructure (14.6%, n = 6), technical support (9.5%, n = 4), and stakeholders' buy-in (0%).

Regarding responsiveness, communication, autonomy, equitability and prompt attention (timeliness of care) demonstrated notable levels of agreement across both the UK and Australia. Participants from the UK reported moderate to high agreement (≥50%) with eight responsiveness factors, listed in descending order: communication (81.3%, n = 52), autonomy (65.6%, n = 42), equitability (62.1%, n = 36), prompt attention (timeliness of care) (59.4%, n = 38), convenience (56.7%, n = 34), choice (selection of FCPS) (53.1%, n = 34), access to social support (51.7%, n = 30), and respect (50.6%, n = 56). Lower levels of agreement (<50%) were reported for other factors, including prompt attention (investigations) (38.7%, n = 24), choice (in-person vs telehealth) (31.3%, n = 20), and confidentiality (5.9%, n = 4). In contrast, Australian respondents reported moderate to high agreement (≥50%) with only four responsiveness items: communication (90%, n = 36), prompt attention (timeliness of care) (66.7%, n = 28), autonomy (60%, n = 24), and equitability (60.5%, n = 23). The remaining items recorded lower levels of agreement (<50%), including prompt attention (investigations) (42.7%, n = 17), convenience (30%, n = 12), confidentiality (23.7%, n = 9), access to social support (23.7%, n = 9), choice (selection of FCPS) (18.8%, n = 6), choice (in-person vs telehealth) (16.7% (n = 5)), respect (7.5%, n = 3). Figs 1A and 1B displays the graphical representation of readiness and responsiveness.

**Effect of timing and country on the types of changes, readiness, and responsiveness.** A two-way ANOVA was performed to assess the effects of timing (within 3 months and >4 months), country (UK and Australia) and their interaction on the type of changes, readiness, and responsiveness (Tables 4 and 5).

Regarding the changes in the access domain, the results indicated no significant main effect for timing (p = 0.154) or for the interaction between timing and country (p = 0.126). There was a significant main effect for country, indicating decreased utilisation of various access methods to FCP services in Australia with mean value ($\bar{x} = -0.371$) compared to the UK ($\bar{x} = 0.415$), (p < 0.001).

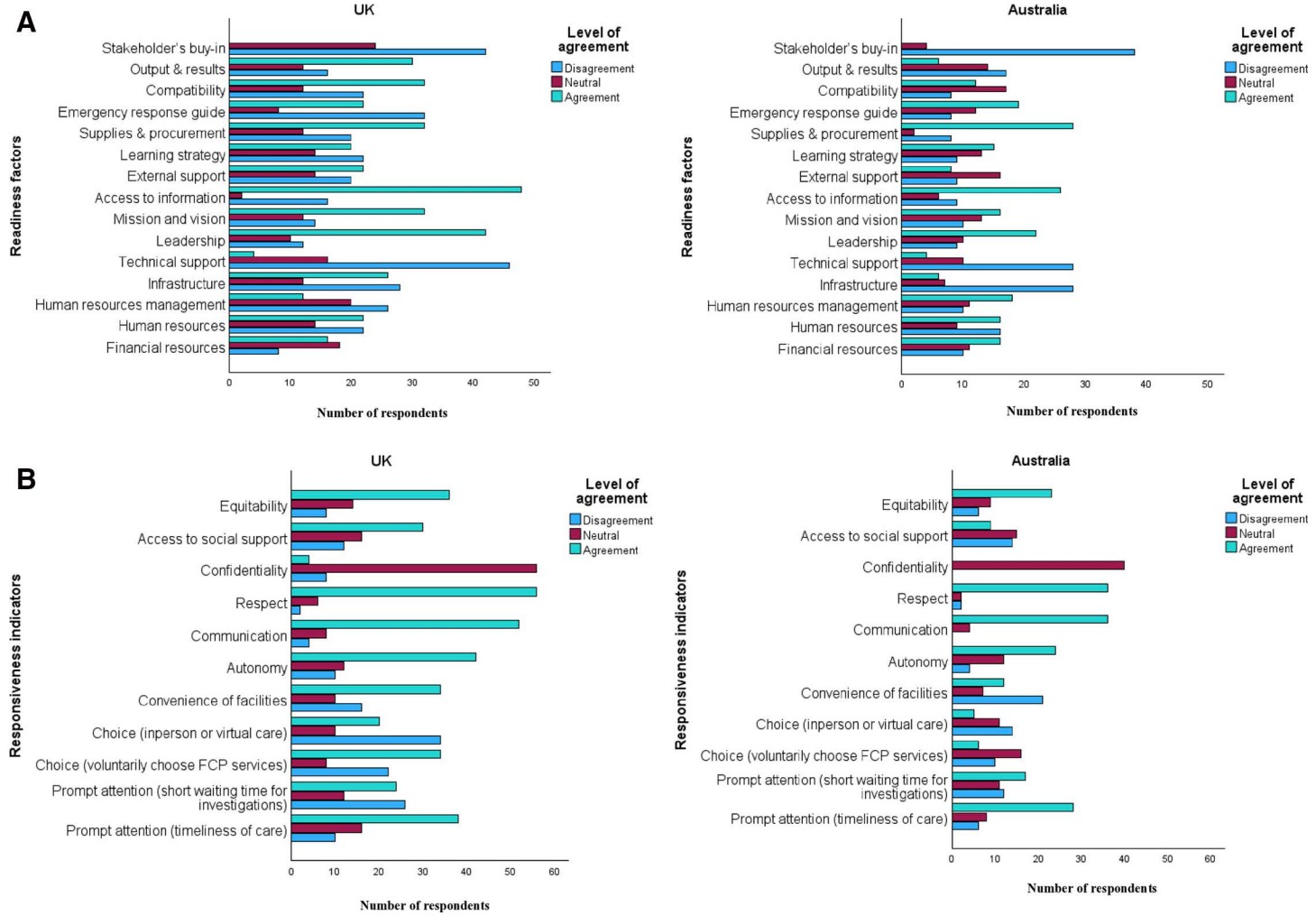

**Fig 1. Readiness and Responsiveness factors.** A. Readiness factors in the UK and Australia. B. Responsiveness factors in the UK and Australia.

In platforms used in delivering MSK care, significant main effects were observed for timing and country, indicating decreased use of diverse delivery platforms within the initial three months ($\bar{x} = -0.009$) compared to four months and beyond ($\bar{x} = 0.247$), (p = 0.014); and decreased change in Australia ($\bar{x} = -0.265$) compared to the UK ($\bar{x} = 0.243$), (p < 0.001). There was no significant interaction between timing and country (p = 0.502).

The presentation of MSK conditions at health facilities showed a significant main effect for both timing and country, indicating a greater decrease in the delivery of services for MSK conditions within the initial three months ($\bar{x} = -0.529$) compared to four months or more ($\bar{x} = -0.174$), (p < 0.001); in Australia ($\bar{x} = -0.778$) compared to the UK ($\bar{x} = -0.211$), (p < 0.001). However, no significant interaction between timing and country (p = 0.398) was observed.

Regarding changes in the interventions provided, there were no significant main effects for timing (p = 0.121) or country (p = 0.125), or interaction between timing and country (p = 0.516). Similarly, the changes occurring in the case load shows no significant main effects for timing (p = 0.938), country (p = 0.413), or interaction between timing and country (p = 0.318).

In terms of changes in administrative and managerial duties, there was no significant main effect for timing (p = 0.627), however, a significant main effect was noted for country, with Australia exhibiting lesser changes ($\bar{x} = 1.62, \ 1.97$)

**Table 4. Descriptive statistics for initial changes by independent variables with 2-way ANOVA.**

| | Within 3 months | | >4 months | | Total | |
|---|---|---|---|---|---|---|
| Variables<br>  Country | $\bar{x}$ | SD | $\bar{x}$ | SD | $\bar{x}$ | SD |
| **Access** | | | | | | |
|   UK | 0.418 | 0.458 | 0.467 | 0.536 | 0.415 | 0.475 |
|   Australia | −0.443 | 0.396 | −1.250 | 0.231 | −0.371 | 0.386 |
|   Total | 0.156 | 0.592 | 0.255 | 0.526 | 0.180 | 0.576 |
| **Platform for delivery** | | | | | | |
|   UK | 0.172 | 0.400 | 0.500 | 0.365 | 0.243 | 0.413 |
|   Australia | −0.310 | 0.589 | −1.212 | 0.454 | −0.265 | 0.561 |
|   Total | −0.009 | 0.531 | 0.247 | 0.503 | 0.049 | 0.534 |
| **MSK patient's presentation** | | | | | | |
|   UK | −0.295 | 0.465 | 0.182 | 0.301 | −0.211 | 0.475 |
|   Australia | −0.850 | 0.254 | −0.530 | 0.433 | −0.459 | 0.502 |
|   Total | −0.529 | 0.476 | −0.174 | 0.515 | −0.459 | 0.502 |
| **Interventions provided** | | | | | | |
|   UK | −0.200 | 0.491 | 0.017 | 0.299 | −0.158 | 0.467 |
|   Australia | −0.288 | 0.422 | −0.198 | 0.201 | −0.265 | 0.378 |
|   Total | −0.234 | 0.465 | 0.086 | 0.274 | −0.202 | 0.434 |
| **Patient's caseload** | | | | | | |
|   UK | 1.969 | 0.603 | 2.099 | 0.646 | 2.002 | 0.613 |
|   Australia | 2.188 | 0.563 | 2.077 | 0.543 | 2.160 | 0.555 |
|   Total | 2.052 | 0.595 | 2.091 | 0.601 | 2.062 | 0.595 |
| **Administrative and managerial duties** | | | | | | |
|   UK | 1.850 | 0.958 | 2.430 | 0.938 | 1.970 | 0.976 |
|   Australia | 1.730 | 0.962 | 1.360 | 0.809 | 1.620 | 0.924 |
|   Total | 1.810 | 0.954 | 1.960 | 1.020 | 1.840 | 0.968 |

$\bar{x}$: Mean value; SD: Standard deviation.

compared to the UK ($\bar{x}$ = 1.97), (p = 0.009). Furthermore, there was a significant interaction between timing and country p = 0.034, with Australia demonstrating significantly less administrative changes in the first three months ($\bar{x}$ = 1.73) than the UK ($\bar{x}$ = 1.85).

For readiness, there were no significant main effects observed for timing (p = 0.144) or country (p = 0.775), and no significant interaction between timing and country (p = 0.721). Also, in terms of responsiveness, there were no significant main effects for timing (p = 0.350) and country (p = 0.910), and no significant interaction between timing and country (p = 0.127).

## Discussion

This study identified that the initial changes in FCPS across the UK and Australia (accounting for both country and service context; PC and ED respectively) commenced within the first three months of the COVID-19 pandemic, within a period now referred to as the first wave [17]. These provider's self-reported changes may have stemmed from the public health policies such as stay at home, social distancing, and telehealth directives [15,17,25,28], triggering an initial response of all healthcare services including the FCPS. This study identified variations in the levels of initial changes reported during the first wave of the pandemic between the UK and Australia, likely due to pre-existing differences in the contexts in which

**Table 5. Effect of timing, country and their interaction with 2-way ANOVA.**

| Variables<br>Source | *Df* | F | $\eta_p^2$ | p value |
|---|---|---|---|---|
| **Access** | | | | |
| Country | 1 | 42.34 | 0.273 | <0.001 |
| Timing | 1 | 2.055 | 0.018 | 0.154 |
| Interaction | 1 | 2.370 | 0.021 | 0.126 |
| Within group error | 113 | | | |
| **Platform for delivery** | | | | |
| Country | 1 | 28.39 | 0.197 | <0.001 |
| Timing | 1 | 6.210 | 0.051 | 0.014 |
| Interaction | 1 | 0.453 | 0.004 | 0.502 |
| Within group error | 116 | | | |
| **MSK patients' presentation** | | | | |
| Country | 1 | 47.09 | 0.304 | <0.001 |
| Timing | 1 | 18.57 | 0.147 | <0.001 |
| Interaction | 1 | 0.719 | 0.007 | 0.398 |
| Within group error | 108 | | | |
| **Interventions provided** | | | | |
| Country | 1 | 2.391 | 0.021 | 0.125 |
| Timing | 1 | 2.439 | 0.021 | 0.121 |
| Interaction | 1 | 0.425 | 0.004 | 0.516 |
| Within group error | 111 | | | |
| **Patients' caseload** | | | | |
| Country | 1 | 0.675 | 0.005 | 0.413 |
| Timing | 1 | 0.006 | 0.000 | 0.938 |
| Interaction | 1 | 1.003 | 0.002 | 0.318 |
| Within group error | 136 | | | |
| **Administrative and managerial duties** | | | | |
| Country | 1 | 7.135 | 0.067 | 0.009 |
| Timing | 1 | 0.237 | 0.002 | 0.627 |
| Interaction | 1 | 4.617 | 0.045 | 0.034 |
| Within group error | 99 | | | |
| **Readiness** | | | | |
| Country | 1 | 0.082 | 0.001 | 0.775 |
| Timing | 1 | 2.168 | 0.018 | 0.144 |
| Interaction | 1 | 0.128 | 0.001 | 0.721 |
| Within group error | 118 | | | |
| **Responsiveness** | | | | |
| Country | 1 | 0.013 | 0.000 | 0.910 |
| Timing | 1 | 0.880 | 0.009 | 0.880 |
| Interaction | 1 | 2.367 | 0.023 | 0.127 |
| Within group error | 100 | | | |

p<0.05.

FCPS operate and the local disease burden [11–13]. For example, while this study showed decreased MSK presentation in the PC, mostly within the first wave of the pandemic, the ED experienced more reduction in MSK presentations, extending beyond the first wave of the pandemic. These variations may also be linked to the increased reordering of MSK condition priorities, a trend reported to be common in ED [25,30,31]. Also, the focus of ED may have shifted to providing care to COVID-19 infected patients, while fear of contracting the virus deterred individuals with other health conditions, such as MSK, from seeking ED services [3,8,9]. In addition, there was decreased community activities and fewer presentations of acute conditions including MSK to the ED [8].

The study findings demonstrated poor adoption of telehealth in the ED FCPS, despite the Australian government's national policy measures instituted to support telehealth advocacy (i.e., access, equity and quality of care), infrastructure and funding during the pandemic [28,29]. This may be because of the urgency, acuteness, and complexity of MSK conditions presented within the ED setting [45], which may cause a distinctive challenge to telehealth use, for instance, managing minor fractures remotely could be more complex. It could also be due to lack of guidelines and standardisation for the use of telehealth as the transition was rapid and seems forced rather than planned, resourced and supported [46]. Interestingly, 20.5% of participants in this study reported an increased use of telehealth in their ED; however, understanding the 'real life' issues that have accompanied implementation of telehealth in the ED fell outside the scope of this study. While a study from Ireland reported the use of telephone consultations for the initial assessment and advice of MSK patients in the ED [32], literature detailing the specific use and application of telehealth technology, particularly for MSK care in ED remains scarce. In contrast, FCPS in the UK PC implemented telephone consultations and telehealth-appropriate interventions [15,17], which improved MSK patient access over time. The rapid adoption of telehealth in PC may also be associated with the type of MSK conditions typically presented, such as chronic MSK conditions [11], and the availability of simple, non-complex technologies like telephones in GP surgeries before the pandemic [47]. A recent study exploring the telehealth use for chronic knee pain in PC revealed that increased availability of resources, such as telehealth educational materials significantly improved the access to MSK care [46].

In terms of readiness for changes in FCPS during the COVID-19 pandemic, participants from both countries reported considerable readiness only in access to information and leadership. There was no agreement on stakeholder buy-in, with respondents expressing disagreement. This reflects the nature of emergency-driven changes, which may have bypassed the usual consideration and consultation of stakeholders (e.g., patients and providers) prior to implementation [36,38]. Interestingly, supplies and procurement achieved the highest agreement of 73.7% among Australian respondents, reflecting the ED context where government-supported resources such as personal protective equipment were available, particularly during the first wave of the pandemic when a surge in care for COVID-infected patients was anticipated [25,30,31]. This availability may have contributed to the higher perceived readiness in this item. However, other readiness factors including human and financial resources, emergency response guidance, and technical support received lower ratings overall. This pattern aligns with findings from other studies, where limitations in these areas hindered healthcare services' capacity to provide continuous care during the pandemic [6,8,21]. Such gaps can slow decision-making, restrict service delivery options, and compromise the ability to maintain continuity of care in rapidly evolving situations [36]. Although this study indicated that changes in FCPS began within the first three months of COVID-19 being declared a global pandemic, the identified readiness limitations may have constrained their overall ability to deliver optimally responsive care, highlighting the need for a proactive and holistic approach to preparedness for future public health emergencies [36].

Considering the changes reported in FCPS, particularly during the initial period of the pandemic, and their level of readiness, to what extent can these services be considered responsive? This study revealed varying levels of responsiveness across different domains. Notably, respondents in both the UK and Australia showed high to moderate agreement on communication, autonomy, equitability, and timeliness of care, with communication achieving the highest agreement (>80%). These findings suggest that many FCPS were perceived as patient-centred and aligned with national COVID-19 policies in both countries, which emphasised clear communication, equity, and business continuity during the pandemic

[15,25]. While the UK demonstrated higher levels of responsiveness in eight items compared to Australia, which reported four out of eleven, likely due to the use of telehealth in PC, other items showed lower responsiveness. This suggests that FCPS efforts, although present, remained insufficient for a prompt and effective emergency response. While some levels of responsiveness were reported overall, the perceived responsiveness occurred alongside a significant reduction in MSK presentations in both PC and ED settings during the initial phase of the pandemic. This reduction may have been influenced by changes in MSK patient behaviour resulting from public health messaging [21–24], rather than reflecting FCPS's inability to provide care, particularly in the ED, where suspension of FCPS was least reported. This may also be partly explained by the early implementation of strict public health measures, which emphasised leadership-driven directives (e.g., prioritising urgent healthcare services, such as care for COVID-19 patients) [5,25], rather than service-level strategies. As with other aspects of this study, these findings may reflect respondents' general perceptions of FCPS's level of responsiveness, rather than their specific experiences during the early stages of the pandemic.

Recognising the diverse contexts in which FCPS operate is crucial, as one-size-fits-all policies may be unsuitable during crises. There is a clear need to strengthen preparedness strategies tailored to these contextual differences, reinforcing existing evidence on the value of context-specific contingency planning [36]. Such planning is essential not only for identifying opportunities for transformative and sustainable change but also for ensuring the continuity of essential services like FCPS during times of crisis. Overall, while this study demonstrates an inadequate level of readiness for responsive change, with only a few readiness items reaching the arbitrary ≥50% agreement cut-off, this threshold still falls short of the level of readiness needed for health services such as FCPS for public health emergencies [1,2]. Achieving higher levels of readiness across all factors will be crucial for the effective and prompt response to and management of future crises.

## Limitations of the study

This study has limitations. First, the definition of "change" in the survey may have lacked clarity. While the pilot study did not indicate this as a problem, it remains uncertain whether these changes reported within the first three months encompassed the suspension of services. Although a separate question addressed service suspension, some participants may have still conflated the concept of change in service delivery with suspension, particularly in the UK, where a high percentage of service suspensions were also reported within the first three months. We recognise that the survey question about changes in service delivery may have introduced variability in how participants interpreted it, particularly regarding whether changes were viewed as formal policy shifts or as shifts in practice, including healthcare-seeking behaviours. However, we intentionally kept the definition broad to reflect the real-world experiences of clinicians, recognising that formal policies were often preceded by formal/informal changes at the service level, especially when considering a significant event such as the COVID-19 pandemic. While there is inevitably some uncertainty around perfect consistency of interpretation and memory, we believe the methods employed were reasonable given the retrospective nature of the study and its aim to capture real-world practice changes during an unprecedented period, as perceived and experienced by the providers.

Second, given the retrospective nature of the survey and the time elapsed since the onset of the COVID-19 pandemic, recall bias is a potential limitation. Although the pilot study and feedback from participants showed that there was adequate recall of experience, also the event timeline prompts, and major public health event (COVID) were used to aid recall [48]. Concerns around consistency and recall accuracy were further addressed through the survey design, which ensured that the selected timeline was consistently carried through all questions, helping to prevent confusion or mixing of timelines. However, this time prompt may have been leading, potentially introducing bias into the responses. Hence, the responses in this study are interpreted as the respondents' accounts of what constituted 'initial' changes for them. We acknowledge that the complexity and evolving nature of service changes during the pandemic could have influenced participants' ability to accurately remember and report the initial changes.

Third, there is ambiguity and variation in the terminology used to describe physiotherapists serving in FCPS across both countries. While "first contact physiotherapist" is widely recognised in the UK and Australia, it is predominantly adopted as such in the UK. In contrast, Australia employs terms such as "primary contact physiotherapist" and "advanced practice physiotherapist," which may involve different roles. However, the survey provided a clear description in its advertisement and introduction using formal descriptions [49,50]. Despite this, confusion about these roles could have contributed to lower survey participation rates, especially in Australia.

Fourth, there were some concerns regarding the study sample size. The sample size constrained the depth of data exploration possible. Finally, responsiveness was assessed through practitioner's perspective only which may not adequately capture the patient's viewpoint. Assessing responsiveness also from the patient's perspective could provide a more comprehensive understanding of FCPS responsiveness during the pandemic. These limitations should be considered when interpreting the findings.

### Recommendations for future studies

Future research should investigate how changes in FCPS during the COVID-19 pandemic impacted MSK patients' health outcomes and satisfaction. Additionally, further study is required to understand how telehealth can be effectively utilised to provide MSK care in FCPS especially within the ED. Evaluating both the effectiveness of readiness strategies, including telehealth interventions, as well as stakeholders' experiences with these approaches is essential to identify best practices for sustaining FCPS during crises. Qualitative methods, such as interviews and focus groups, could offer valuable insights into the experiences, challenges, and facilitators involved. It is also valuable to conduct an in-depth comparison of readiness and responsiveness across various healthcare systems and policy environments to better understand how structural and system level factors may impact FCPS service delivery. Additionally, a longitudinal approach to research can be employed to track how FCPS readiness and responsiveness evolves over time.

### Conclusion

Changes in FCPS were reported to have commenced within the first three months following the World Health Organisation's declaration of the COVID-19 pandemic, in both the UK and Australia. These changes likely varied due to differences in the contexts in which FCPS operate [10,11], potentially influenced by local policies, capabilities, service needs, supply and demand during the COVID-19 pandemic [6,15–18]. Even though changes were reported to have occurred within the first three months in both contexts, both demonstrated similar levels of inadequate readiness for rapid changes in service delivery. However, respondents from the UK FCPS reported agreement on more responsiveness items than Australia, potentially supported by the rapid implementation of telehealth in UK PC for MSK patients. While this study used an arbitrary cut-off of ≥50% to indicate moderate to high agreement in readiness and responsiveness, achieving high agreement across all readiness factors will be essential to prepare FCPS for more responsive future crises. This underscores the need to learn from past experiences, invest in the wins and support FCPS to build resilience in their operations to rapidly assess demand, match supply and create flexible and agile plans to prevent or minimise service disruptions in future crises.

### Supporting information

**S1 Checklist. Checklist for reporting results of internet e-surveys (CHERRIES) for FCPS survey.**
(DOCX)

**S2 Survey. FCPS survey, UK and Australia.**
(PDF)

## Author contributions

**Conceptualization:** Oluwatoyin Adenike Adeniji, Nicola Carey, Theopisti Chrysanthaki.

**Data curation:** Oluwatoyin Adenike Adeniji.

**Formal analysis:** Oluwatoyin Adenike Adeniji.

**Funding acquisition:** Oluwatoyin Adenike Adeniji.

**Investigation:** Oluwatoyin Adenike Adeniji.

**Methodology:** Oluwatoyin Adenike Adeniji, Karen Stenner, Nicola Carey, Theopisti Chrysanthaki.

**Project administration:** Oluwatoyin Adenike Adeniji.

**Resources:** Oluwatoyin Adenike Adeniji.

**Supervision:** Evangelos Pappas, Karen Stenner, Victoria Traynor, Nicola Carey, Theopisti Chrysanthaki.

**Validation:** Oluwatoyin Adenike Adeniji, Evangelos Pappas, Karen Stenner, Victoria Traynor, Nicola Carey, Theopisti Chrysanthaki.

**Visualization:** Oluwatoyin Adenike Adeniji.

**Writing – original draft:** Oluwatoyin Adenike Adeniji.

**Writing – review & editing:** Oluwatoyin Adenike Adeniji, Evangelos Pappas, Karen Stenner, Victoria Traynor, Nicola Carey, Theopisti Chrysanthaki.

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
