## [Decision Letter · Decision Letter 0]

2 Apr 2025

Dear  Adeniji,

Thank you for submitting your manuscript to PLOS ONE. After careful consideration, we feel that it has merit but does not fully meet PLOS ONE’s publication criteria as it currently stands. Therefore, we invite you to submit a revised version of the manuscript that addresses the points raised during the review process.

We look forward to receiving your revised manuscript.

Kind regards,

Opeyemi O Babatunde, Ph.D., MPh., B.Physio

Academic Editor

PLOS ONE

Journal Requirements:

4. Please note that your Data Availability Statement is currently missing [the repository name and/or the DOI/accession number of each dataset OR a direct link to access each database]. If your manuscript is accepted for publication, you will be asked to provide these details on a very short timeline. We therefore suggest that you provide this information now, though we will not hold up the peer review process if you are unable.

5. Please include a caption for figure 1.

Reviewers' comments:

Reviewer's Responses to Questions

**Comments to the Author**

1. Is the manuscript technically sound, and do the data support the conclusions?

Reviewer #1: No

Reviewer #2: Yes

2. Has the statistical analysis been performed appropriately and rigorously?

Reviewer #1: Yes

Reviewer #2: Yes

3. Have the authors made all data underlying the findings in their manuscript fully available?

Reviewer #1: Yes

Reviewer #2: Yes

4. Is the manuscript presented in an intelligible fashion and written in standard English?

Reviewer #1: Yes

Reviewer #2: Yes

Reviewer #1: I generally found that the conclusion sections in both the abstract and main text did not accurately reflect the findings of the analysis outlined in the Results and Discussion section, and at times included statements that appear to be unsubstantiated by the study. I would suggest that these sections are revised. Please find my specific comments below.

1. Weathering the storm of COVID-19 pandemic in UK and Australia First Contact Physiotherapy

Services: A cross-contextual survey investigating the timing and type of rapidly implemented

changes

Could the authors please clarify how they defined “rapidly implemented changes”, and in what way the questionnaire ensured that respondents only focused on these? Note that although one of the main findings of the survey is that the timing of changes implementation was mostly rapid (within 3 months), this does not mean that the survey was investigating rapidly implemented changes.

2. Aim: This study aims to explore the type, and the timing of changes implemented in FCPS during the COVID-19 pandemic in the UK and Australia.

The manuscript frequently references "changes implemented" or "adaptations" in service provision (including in the title). I have two concerns in this regard:

- The changes considered in the survey (Qs 10 to 26) do not differentiate between those formally introduced as policy changes by healthcare providers and unintended changes that may have arisen due to other factors. Such factors include shifts in population needs (e.g., linked to lifestyle changes associated with lockdowns), alterations in healthcare-seeking behaviours, or modifications in healthcare providers’ working patterns that were demand- rather than policy-driven. The authors should clarify throughout the manuscript that the study aimed to document perceived changes in service provision, without implying that these changes were necessarily the result of “adaptations” by healthcare providers. In this particular sentence, I would also clarify that these changes were “self-reported by FCPS providers”.

- The study only considered *initial* changes to FCPS (see point 5). This should be clarified throughout the manuscript.

3. From the questionnaire

“For some, changes occurred in the first three months of the COVID 19 pandemic while it took more months for others.”

One of the main findings of this study appears to be that the initial changes to MSK service delivery were mostly implemented during the first three months of the global pandemic. I therefore find it concerning that the introduction to the questionnaire appears to be highlighting “in the first three months” as a plausible answer to the timing questions Qs 5 and 7. Although “more months” is mentioned in the same sentence, it aggregates all other possible answers to these questions into one category, potentially emphasising the first timeframe more strongly. This is concerning given that respondents are being asked to recall events that occurred three years prior, making their recollections susceptible to suggestion. How do the authors justify the need to include this sentence in the questionnaire’s introduction? This potential source of bias should be acknowledged in the discussion.

4. From the questionnaire “The following set of questions will explore changes to access, mode of delivery, condition managed, interventions provided, patient case load, and administrative duties in your FCP service. The responses you provide should be based on the first changes that occurred in your FCP service delivery for MSK patients during the pandemic”

Participants were asked to describe the “first changes” that occurred in service delivery. This phrasing is ambiguous, as it is unclear:

- Whether these changes should be interpreted as formal policy implementations or simply as observations of shifts in practice (e.g., changes in healthcare-seeking behaviour);

- Which period of time respondents should have considered as representing the “first changes.” Does this refer to any period until another change was introduced (or observed) in service provision? How significant must a subsequent change have been to delineate the timeframe?

- How it relates to previous questions on service suspension;

Given these ambiguities, how confident are the authors that participants would have interpreted these instructions in a consistent manner? And how certain are they that respondents could accurately recall time variations in service provision three years later, particularly considering the likelihood of multiple policy changes throughout the pandemic on timescales of weeks/months?

5. This study aims to explore the type and the timing …

I have two concerns regarding the use of the term “timing” in this context. Firstly, the study focuses exclusively on the initial implementation of changes to service provision and does not account for how these changes may have evolved throughout the pandemic. This limitation means the study cannot determine whether more significant changes occurred at later stages of the pandemic, or how long those initial changes lasted (see point 4). Since the phrase “timing of changes” typically implies not only their onset but also their progression over time, its use here may be misleading. I suggest making it clear that the study concerns only the onset of changes.

Secondly, I am concerned about the use of a “universal” time frame to calibrate the timing of changes. As noted in the manuscript, “Each country responded differently to the COVID-19 pandemic,” and “lockdown measures were implemented across Australian states with varied timing.” Furthermore, the timing and severity of pandemic waves differed significantly between countries, regions, and states. How do the authors justify using the declaration of COVID-19 as a global pandemic (by whom? Please add) as a uniform reference point across the UK and Australia, considering these contextual differences?

6. “An international cross-sectional study was conducted, using an online survey tool. Data were

collected from January-April 2023 from physiotherapists managing MSK patients in PC and ED across the UK and Australia, respectively”.

It should be made clear that the data was collected from physiotherapists who fulfilled the study’s eligibility criteria, which stated that only responses from providers who recalled changes in MSK service delivery would be considered.

7. One hundred and fifty-three participants were included in the analysis with 64.1% based in the UK and 35.9% in Australia

This should be in the methodology, not the results.

8. The main age group were 35-44 years (42.2%), with 59.3% females

The age and gender of the provider was not shown to be significant in any part of the analysis. Why is it included as a main result in the abstract?

9. 75.7% of the changes in the delivery…

The questionnaire only asked participants when the first changes took place, so this sentence should read "75.7% of initial changes" or similar.

10. “75.7% of the changes in the delivery of FCPS mostly occurred during the initial three months

of COVID-19’s declaration as a global pandemic”

What does “mostly” mean in this sentence?

11. The timing and country had a mixed effect on the variables of change.

Please report the most significant results in the abstract.

12. Adaptations in FCPS during the COVID-19 pandemic varied across countries

This study only considered two countries, and therefore does not provide enough statistical evidence to support this statement.

Furthermore, due to the complete overlap between the country and the type of service delivery setting considered (PC settings in the UK and ED settings in Australia), it is impossible to determine whether similar delivery settings in the two countries experienced different or similar changes in FCPS provision. To substantiate such a claim, the authors would need to collect additional data from PC settings in Australia and ED settings in the UK and account for these variables in their analysis.

With regards to the use of the term “adaptations” in this context, please refer to comment 2.

13. This variation underscores the need for timely and context-specific improvement strategies

Variation in observed changes between two different countries in itself doesn’t underscore a) the need for improvements, or b) the need for these improvements to be “context specific” (observing variation does not rule out the possibility that adaptations introduced in country A would have led to better outcomes in country B had they been applied there too). Furthermore, the analysis of the survey data did not investigate the link between the type of changes observed and the quality of service delivered, and therefore cannot provide any conclusions as to what type of changes - context specific or not - led to better outcomes. Please rewrite the conclusions to better reflect the findings reported in the results section.

14. The UK reported 59.2% (n=42) increased use of the triaging process

In this and many other instances where results are reported (e.g. In terms of intervention provided during the pandemic, self-management increased more in the UK 86.8% (n=59), compared to Australia 44.7%) the imprecise phrasing makes the meaning of the sentence open for misinterpretation: As is, the sentence may be incorrectly interpreted as suggesting that the UK reported a 59.2% increase in the triaging process, whereas it should read “Among UK-based respondents, 59.2% observed an increased use of the triaging process”. Please correct this throughout the manuscript, and also in the Tables.

15. The COVID-19 pandemic led to changes in FCPS within the first three months of its declaration as a global pandemic in both the UK and Australia.

The “eligibility criteria” used to select the 153 survey participants considered in this analysis means this sample cannot be used to determine whether any statistically significant changes to MSK service provision occurred during the COVID-19 pandemic in either country. This sentence, therefore, is unsubstantiated unless a reference is provided or further analysis conducted.

The survey collected raw data (which was excluded from the sample of data analysed) from respondents who reported observing no changes in service provision during the pandemic (20.3% overall). If analysed, this data could provide some evidence in support of this statement. However, the title of the survey itself (“Exploring changes to first contact physiotherapy services for musculoskeletal patients during COVID-19 Pandemic”) and stated objectives is very likely to have introduced a bias in the collection of responders, favouring participation from individuals who indeed observed changes in service provision. The authors should either a) conduct an analysis on the statistical significance of the claim that the COVID19 pandemic led to changes in FCPS from their complete raw data sample while providing a satisfactory explanation for why the above-mentioned bias is not a cause for concern, b) provide a reference for this sentence, or c) remove this sentence.

16. These changes varied depending on contextual factors influenced by policies, needs, supply and demands during the pandemic

This study did not consider the role of policy implementation, nor collect statistics on the needs in the local population, or on the “demand” during the pandemic. This statement therefore is not substantiated by the study.

17. For instance, patients continued to have access to the FCPS due to the availability of telehealth in the PC setting, while the ED saw a significant reduction in MSK patient presentations, as it primarily continued with traditional face-to-face services based on the need for hands-on acute care.

This statement seems to be in direct contradiction with the earlier statement that “Among UK respondents, 60.0% (n=53) reported suspensions, compared to only 9.3% (n=5) of respondents from Australia”. How do you reconcile the two statements?

18. This underscores the necessity for the development of context tailored implementation

Since the relationship between observed changes (some of which may have been beyond the control of providers and therefore fall beyond the scope of ‘implementation’), “context” (how is context defined?), and outcomes haven't been investigated in this analysis, this statement is unsubstantiated by this study.

Reviewer #2: This is an interesting topic and author write it well but still need some gramatical corrections. Valid stat analysis is done. authors has makrked all the valuable points in manuscript. Despite some corrections are required which are mention below:

introduction need to be deined more explaining about the topic

discussion also need some minor corrections

**Do you want your identity to be public for this peer review?** For information about this choice, including consent withdrawal, please see our Privacy Policy

Reviewer #1: **Yes:**  Margherita Molaro

Reviewer #2: No

---

## [Author Response · Author response to Decision Letter 1]

13 May 2025

Responses to reviewers and editor

We sincerely appreciate the editor and reviewers for their time, commitment, and effort in helping to strengthen our paper. We have addressed the concerns raised below. The main document is marked with tracked changes. Also, a clean copy has been attached.

REVIEWER 1

I generally found that the conclusion sections in both the abstract and main text did not accurately reflect the findings of the analysis outlined in the Results and Discussion section, and at times included statements that appear to be unsubstantiated by the study. I would suggest that these sections are revised. Please find my specific comments below.

1. Weathering the storm of COVID-19 pandemic in UK and Australia First Contact Physiotherapy

Services: A cross-contextual survey investigating the timing and type of rapidly implemented

changes

Could the authors please clarify how they defined “rapidly implemented changes”, and in what way the questionnaire ensured that respondents only focused on these? Note that although one of the main findings of the survey is that the timing of changes implementation was mostly rapid (within 3 months), this does not mean that the survey was investigating rapidly implemented changes.

Response: The title has been revised for consistency and to more accurately reflect the study’s focus. The term "rapidly implemented" was originally intended to capture the speed at which FCPS responded to the pandemic. We used the definitions from a recent systematic review (Adeniji et al., 2025) that reported that most healthcare service adaptations during the COVID-19 pandemic were implemented within the first three months following the WHO’s declaration that would be considered a rapid response during a crisis. Although the term "rapidly implemented" has been removed from the title and throughout the paper for consistency, this study still uses the timeline identified in the literature as a reference point for categorising the timing of changes. We ensured that the respondents only focused on the initial changes by including questions related to the timing of the changes in the questionnaire (<3 months and 3-6 months, etc).

2. Aim: This study aims to explore the type, and the timing of changes implemented in FCPS during the COVID-19 pandemic in the UK and Australia. The manuscript frequently references "changes implemented" or "adaptations" in service provision (including in the title). I have two concerns in this regard: - The changes considered in the survey (Qs 10 to 26) do not differentiate between those formally introduced as policy changes by healthcare providers and unintended changes that may have arisen due to other factors. Such factors include shifts in population needs (e.g., linked to lifestyle changes associated with lockdowns), alterations in healthcare-seeking behaviours, or modifications in healthcare providers’ working patterns that were demand- rather than policy-driven. The authors should clarify throughout the manuscript that the study aimed to document perceived changes in service provision, without implying that these changes were necessarily the result of “adaptations” by healthcare providers. In this particular sentence, I would also clarify that these changes were “self-reported by FCPS providers”.- The study only considered *initial* changes to FCPS (see point 5). This should be clarified throughout the manuscript.

Response: Thank you for making this distinction. We have now revised the paper throughout to talk about “initial”, “self-reported” and “perceived” changes. This has also been established in the introduction as below:

“Therefore, this study aims to explore changes in FCPS during the COVID-19 pandemic in the UK and Australia, with a focus on the initial changes. This approach helps to gauge the speed of FCPS response and readiness based on physiotherapists’ self-reported experiences in the FCPS role”.

3.From the questionnaire “For some, changes occurred in the first three months of the COVID 19 pandemic while it took more months for others.” One of the main findings of this study appears to be that the initial changes to MSK service delivery were mostly implemented during the first three months of the global pandemic. I therefore find it concerning that the introduction to the questionnaire appears to be highlighting “in the first three months” as a plausible answer to the timing questions Qs 5 and 7. Although “more months” is mentioned in the same sentence, it aggregates all other possible answers to these questions into one category, potentially emphasising the first timeframe more strongly. This is concerning given that respondents are being asked to recall events that occurred three years prior, making their recollections susceptible to suggestion. How do the authors justify the need to include this sentence in the questionnaire’s introduction? This potential source of bias should be acknowledged in the discussion.

Response: The survey used event timelines to prompt respondents' memories, to minimise recall bias, helping participants to contextualise their perspectives and experiences within a specific timeframe (Moreno-Serra et al., 2022). We acknowledge that using a time prompt may introduce potential leading, resulting in bias, this has now been addressed in the limitations section.

4. From the questionnaire “The following set of questions will explore changes to access, mode of delivery, condition managed, interventions provided, patient case load, and administrative duties in your FCP service. The responses you provide should be based on the first changes that occurred in your FCP service delivery for MSK patients during the pandemic” Participants were asked to describe the “first changes” that occurred in service delivery. This phrasing is ambiguous, as it is unclear:- Whether these changes should be interpreted as formal policy implementations or simply as observations of shifts in practice (e.g., changes in healthcare-seeking behaviour);- Which period of time respondents should have considered as representing the “first changes.” Does this refer to any period until another change was introduced (or observed) in service provision? How significant must a subsequent change have been to delineate the timeframe? - How it relates to previous questions on service suspension; Given these ambiguities, how confident are the authors that participants would have interpreted these instructions in a consistent manner? And how certain are they that respondents could accurately recall time variations in service provision three years later, particularly considering the likelihood of multiple policy changes throughout the pandemic on timescales of weeks/months?

Response: Thank you for this thoughtful observation. We acknowledge that the phrasing regarding the "first changes" could have introduced variability in interpretation among respondents. To address these concerns:

Formal vs. observed changes: Given the rapid and dynamic nature of the pandemic response, we intentionally kept the definition broad to reflect the real-world experiences of clinicians, recognising that formal policies were sometimes preceded by formal/informal changes at the service level, especially when considering a significant event such as the COVID-19 pandemic.

Timeframe and subsequent changes: Participants were prompted to focus on the earliest significant change they experienced in service delivery, rather than later developments. While we recognise that subsequent changes could have blurred the memory of initial changes, the survey specifically asked about the first discernible changes to mitigate this issue.

Relationship to service suspension questions: Where questions about service suspension were asked separately, they aimed to determine if services ceased altogether, while the "first changes" questions referred to modifications in how services were delivered (e.g., move to telehealth, triage adjustments).

Consistency and recall accuracy: We acknowledge the potential for variability and recall bias, particularly given the time elapsed since the onset of the pandemic. This was noted as a limitation in our study. However, to help mitigate this, we used prominent public events (e.g., WHO's pandemic declaration) as temporal anchors and included timeline prompts within the survey to aid recall and provide a consistent reference point across participants. In addition, the survey was designed so that the selected timeframe was consistently carried through all questions, helping to prevent confusion or mixing of timelines.

While there is inevitably some uncertainty around consistency of interpretation and memory, we believe the methods employed were reasonable given the retrospective nature of the study and the aim to capture real-world practice changes during an unprecedented period. In addition, even though we have anchored the development of the survey on the latest evidence and best practice, we acknowledge that recall bias, variable definitions and the potential for different interpretations are inherent in survey research. These limitations have been acknowledged.

5. This study aims to explore the type and the timing. I have two concerns regarding the use of the term “timing” in this context. Firstly, the study focuses exclusively on the initial implementation of changes to service provision and does not account for how these changes may have evolved throughout the pandemic. This limitation means the study cannot determine whether more significant changes occurred at later stages of the pandemic, or how long those initial changes lasted (see point 4). Since the phrase “timing of changes” typically implies not only their onset but also their progression over time, its use here may be misleading. I suggest making it clear that the study concerns only the onset of changes.

Response: The title has been rephrased to accurately reflect the scope of the study. We acknowledge the importance of assessing longitudinal changes throughout the course of the pandemic, this is out of scope of this project. The study aim has been revised to clarify the focus on the initial changes that may have been introduced in practice. The revision helps eliminate any misleading interpretation and better captures the study’s real intent. In addition, the paper has been revised throughout to more accurately reflect the findings, particularly in clarifying the initial onset of changes as perceived by providers, where applicable.

Secondly, I am concerned about the use of a “universal” time frame to calibrate the timing of changes. As noted in the manuscript, “Each country responded differently to the COVID-19 pandemic,” and “lockdown measures were implemented across Australian states with varied timing.” Furthermore, the timing and severity of pandemic waves differed significantly between countries, regions, and states. How do the authors justify using the declaration of COVID-19 as a global pandemic (by whom? Please add) as a uniform reference point across the UK and Australia, considering these contextual differences?

Response: Thank you for this thoughtful observation. We fully acknowledge the variability in the timing and severity of COVID-19 waves and responses across countries, regions, and even states. However, the decision to use the World Health Organisation’s (WHO) declaration of COVID-19 as a global pandemic (March 11, 2020) as a reference point was based on its symbolic and practical value as a widely recognised turning point in public and institutional awareness. This declaration marked a critical shift in global and national responses, often triggering more formalised actions, including lockdowns, resource reallocation, and shifts in healthcare delivery. Although the implementation of these measures varied, insights from our preliminary one-to-one consultations during the study's conceptualisation phase, along with existing literature, indicated that the WHO declaration served as a common temporal anchor in both professional and public consciousness. Using this reference point also enhances comparability across diverse settings, particularly in an international study where more context-specific time frames may complicate analysis and interpretation. We have now clarified this rationale (under methods) and added who made the declaration in the revised manuscript. The contextual differences between countries and settings under study are acknowledged in the introduction.

6. “An international cross-sectional study was conducted, using an online survey tool. Data were

collected from January-April 2023 from physiotherapists managing MSK patients in PC and ED across the UK and Australia, respectively”. It should be made clear that the data was collected from physiotherapists who fulfilled the study’s eligibility criteria, which stated that only responses from providers who recalled changes in MSK service delivery would be considered.

Response: Information on eligibility has now been added to the abstract (methods) and manuscript sampling section as shown in query no 13.

7. One hundred and fifty-three participants were included in the analysis with 64.1% based in the UK and 35.9% in Australia. This should be in the methodology, not the results.

Response: This has now been removed from the abstract results and moved to the abstract methods section as shown in query no 13.

8. The main age group were 35-44 years (42.2%), with 59.3% females. The age and gender of the provider was not shown to be significant in any part of the analysis. Why is it included as a main result in the abstract?

Response: Thank you for your comment. This has now been removed as shown in query no 13.

9. 75.7% of the changes in the delivery…The questionnaire only asked participants when the first changes took place, so this sentence should read "75.7% of initial changes" or similar.

Response: This has now been revised to “initial changes” as shown in query no 13

10. “75.7% of the changes in the delivery of FCPS mostly occurred during the initial three months

of COVID-19’s declaration as a global pandemic” What does “mostly” mean in this sentence?

Response: This was a typo error, and “mostly” has now been removed as shown in query no 13

11. The timing and country had a mixed effect on the variables of change.

Please report the most significant results in the abstract.

Response: The abstract has been re-written and significant results have now been added as shown in query no 13.

12. Adaptations in FCPS during the COVID-19 pandemic varied across countries. This study only considered two countries and therefore does not provide enough statistical evidence to support this statement. Furthermore, due to the complete overlap between the country and the type of service delivery setting considered (PC settings in the UK and ED settings in Australia), it is impossible to determine whether similar delivery settings in the two countries experienced different or similar changes in FCPS provision. To substantiate such a claim, the authors would need to collect additional data from PC settings in Australia and ED settings in the UK and account for. these variables in their analysis. With regards to the use of the term “adaptations” in this context, please refer to comment 2.

Response: The term "adaptation" has been replaced with "changes" in the abstract and throughout the study to ensure consistency and clarity as shown in query no 13. The abstract has now been re-written, and we revised the paper throughout to reflect that only two countries were included and that due to the nature of the FCPS in the UK and Australia the settings were different as shown in query no 13.

13. This variation underscores the need for timely and context-specific improvement strategies

Variation in observed changes between two different countries in itself doesn’t underscore a) the need for improvements, or b) the need for these improvements to be “context specific” (observing variation does not rule out the possibility that adaptations introduced in country A would have led to better outcomes in country B had they been applied there too). Furthermore, the analysis of the survey data did not investigate the lin

---

## [Decision Letter · Decision Letter 1]

1 Jul 2025

Dear Dr. Adeniji,

Thank you for submitting your manuscript to PLOS ONE. After careful consideration, we feel that it has merit but does not fully meet PLOS ONE’s publication criteria as it currently stands. Therefore, we invite you to submit a revised version of the manuscript that addresses the points raised during the review process.

publication criteria  and not, for example, on novelty or perceived impact.

We look forward to receiving your revised manuscript.

Kind regards,

Opeyemi O Babatunde, Ph.D., MPh., B.Physio

Academic Editor

PLOS ONE

Journal Requirements:

Additional Editor Comments:

Reviewers' comments:

Reviewer's Responses to Questions

**Comments to the Author**

Reviewer #1:

Reviewer #3: (No Response)

Reviewer #4: All comments have been addressed

Reviewer #5: (No Response)

Reviewer #6: (No Response)

2. Is the manuscript technically sound, and do the data support the conclusions?

Reviewer #1: No

Reviewer #3: Partly

Reviewer #4: Yes

Reviewer #5: Partly

Reviewer #6: Yes

3. Has the statistical analysis been performed appropriately and rigorously?

Reviewer #1: Yes

Reviewer #3: Yes

Reviewer #4: Yes

Reviewer #5: (No Response)

Reviewer #6: Yes

4. Have the authors made all data underlying the findings in their manuscript fully available?

Reviewer #1: Yes

Reviewer #3: No

Reviewer #4: Yes

Reviewer #5: Yes

Reviewer #6: Yes

5. Is the manuscript presented in an intelligible fashion and written in standard English?

Reviewer #1: Yes

Reviewer #3: No

Reviewer #4: Yes

Reviewer #5: Yes

Reviewer #6: Yes

Reviewer #1: The authors have comprehensively addressed the points raised in my first review. I believe currently there is only one major outstanding issue that should be addressed by the authors. I also include a few minor comments.

Major comment:

The manuscript repeatedly states (in the abstract, results, and conclusions section) that “both [countries] demonstrated similar levels of inadequate readiness for rapid changes in service delivery and moderate levels of responsiveness to the healthcare needs of MSK patients”. I believe at present the analysis has not sufficiently demonstrated either claim.

With regards to the claim of “moderate levels of responsiveness”: The study should more clearly define the use of the term “moderate”. In the text, however, this is discussed as being indicative of an issue with responsiveness (e.g. “The moderate level of FCPS responsiveness could be attributed to *inadequate* readiness”, “the changes reported in FCPS in the UK and Australia during the pandemic generally reflect the experiences of changes reported in healthcare service [..] exposed the fragility of the current health systems and their unpreparedness to deal with the turbulence of the unexpected”).

However, from what I can see from Fig. 1B, in the UK only in the case of “Choice (in person or virtual)” did a majority of respondents disagree with the statement “I believe my FCP services met the patient's expectations and needs”. In the case of Australia, in none of the criteria a majority of respondents disagreed with that statement.

If only in a minority of cases were issues identified with the level of care received by the patients, why is this indicative of an issue with responsiveness? In what way does it reflect “the fragility of the current health systems and their unpreparedness to deal with the turbulence of the unexpected”?

With regards to the claim of “inadequate readiness”: It is unclear how the authors determined that both settings demonstrated levels of inadequate readiness; in particular:

Based on Fig 1A, respondents were asked to express their opinion on readiness in 15 different areas. However the results section only reports answers to five (access to information, effective leadership, stakeholder buy-in, tech support, and infrastructure); why was only this subset considered? Why were areas as important as “Outputs & Results” not included?

The criteria used to determine whether there is overall agreement with a particular statement are unclear. Based on my understanding, the survey asked respondents whether they “agreed,” “disagreed,” or were “neutral” regarding readiness in various areas. When reporting overall agreement with readiness, the authors appear to use a majority rule—that is, they consider respondents to be in agreement if more than 50% selected “agree”. However, it seems the authors apply a different standard when interpreting disagreement.

Specifically, they appear to treat disagreement with readiness as equivalent to agreement with a lack of readiness (this in itself should be explicitly stated in the text as an assumption), but without requiring a majority for overall interpretation. For example, in support of the statement that “participants agreed there was a lack of readiness for change in terms of infrastructure,” they cite agreement levels of 42.4% (UK) and 39.0% (Australia)—both below the 50% threshold. By contrast, in other areas such as access to information or effective leadership, the authors emphasize whether a majority agreed, suggesting inconsistency in their criteria for interpreting overall agreement levels. I believe their reasoning is that, in the case of the UK, more respondents selected “disagree” than selected “agree” or “neutral” individually. However, this still means that only a minority (<50%) of participants explicitly disagreed with the statement of readiness. Therefore, I do not think the authors can reasonably claim that respondents “agreed that there was lack of readiness for change in terms of infrastructure” based on this result.

In the case of Australia, however, I believe the authors have made a mistake when quoting 39% in disagreement with readiness of infrastructure, based on Fig 1A. Please clarify the discrepancy.

Based on the subset of 5 areas considered, it would appear that only in 2 out of 5 cases a majority of respondents thought that their FCPS was not ready for rapidly implemented changes. In what way does this support the statement of “inadequate readiness for rapid changes in services”?

To establish whether this claim can be supported, authors should 1) use consistent criteria of majority, and 2) summarise all areas investigated in the survey (especially “results and outcomes”) or motivate why only 5/15 are relevant.

Overall, I believe the survey results are more nuanced—and at times contradictory to existing literature—than currently acknowledged. The conclusions should reflect this complexity, not just for the sake of accuracy, but also because it adds to the interest and value of the study.

Minor comments:

- Suggest using “reported changes” in the title

- Fig 1 axes should be number of respondents, not frequency.

- Throughout the manuscript, clarify that when analysing responses as a function of country, this is really a function of country + type of service (ED and PC) because of the degeneracy with which the survey was constructed. Maybe this combination could be redefined as “setting”.

Reviewer #3: Thanks for your effort to cover the comments but In most cases, the answers and edits made did not cover the reviewer's comments.

Reviewer #4: Authors carefully responded to Reviewer 1's remarks by revising the title, abstract, introduction, and discussion for clearer definition of the scope of the study on first, perceived, and self-reported changes early in COVID-19 in First Contact Physiotherapy Services (FCPS). They acknowledged limitations such as recall bias, ambiguity in definitions, and sample representativeness. Technical terms such as "adaptations", "timing" were also clarified to prevent misinterpretation. Findings and tables were rewritten to ensure accuracy and unsubstantiated claims were removed and made more specific. Structural and grammatical suggestions for introduction and discussion were applied. Editorial requests were accommodated, such as ethics clarification, participant consent information, data availability via a DOI, and format changes according to PLOS ONE guidelines. Overall, the authors thoroughly revised with substantive changes that sufficiently resolved the reviewer's and editor's concerns and made the manuscript more transparent, consistent, and scientifically valid. Revisions reflect a strong effort to align the content with journal standards and reviewer expectations.

Reviewer #5: The definitions of "rapidly implemented changes" and the vagueness of the survey questions could cause respondents to perceive them differently. Concerns around recall bias are further raised by the use of self-reported data. The findings are not well represented in the conclusion, and the authors ought to make a clearer contrast between observed changes in practice and official policy changes. The approach, a cross-sectional survey using physiotherapists' self-reported data, is suitable for documenting perceived changes. However, issues with recall bias and the imprecise definition of "first alterations" could compromise the validity of the results. Overall, even though the goal is pertinent, the method's drawbacks must be carefully considered. Resolving these problems could improve the validity and significance of the study.

Reviewer #6: Thank you for the invitation to review the manuscript titled “Weathering the storm of COVID-19 pandemic: A cross-sectional survey of changes in first contact physiotherapy services in the UK and Australia”

The manuscript is concise and scientifically written using recent literature

I have the following comments to offer in order to improve the quality of the study for use especially during pandemic period to enhance healthcare services in the two regions and globally

Reviewer: Adamu Ahmad Adamu PT PhD

Comments:

1. Authors should give adequate explanation to changes in service delivery

2. Conclusion in the abstract should highlight the effect on changes in service delivery in relation to future pandemic

3. What are the initial changes noticed in UK and Australia during the pandemic?

4. How did the authors ensured the participants were able to recall changes in their practice during the pandemic, was any instrument used to ascertain cognitive function?

5. What is the educational level of the participants?

6. The sample size is a bit low to make conclusion based on the findings

7. Explain the experiences and qualifications of the 12 participants drawn for the pilot test of the study instrument

8. In Table 1, the first column should have a heading like item or domain for clarity

9. Recommendation for future study should be incorporated to address gaps in the existing study

10. Any conflict of interest resolved by the authors?

**Do you want your identity to be public for this peer review?** For information about this choice, including consent withdrawal, please see our Privacy Policy

Reviewer #1: **Yes:**  Margherita Molaro

Reviewer #3: No

Reviewer #4: **Yes:**  Fawaz Alrasheedi

Reviewer #5: **Yes:**  Ahmed Ibrahim Ahmed Al Kharusi

Reviewer #6: **Yes:**  Adamu Ahmad Adamu

---

## [Author Response · Author response to Decision Letter 2]

15 Aug 2025

Response to reviewers and editor

We sincerely thank the reviewers and editor for their thoughtful and constructive feedback. The manuscript has been carefully revised in response to the comments, with all updated sections clearly highlighted in yellow for ease of reference. Detailed responses to each point raised are provided below.

Reviewer #1: The manuscript repeatedly states (in the abstract, results, and conclusions section) that “both [countries] demonstrated similar levels of inadequate readiness for rapid changes in service delivery and moderate levels of responsiveness to the healthcare needs of MSK patients”. I believe at present the analysis has not sufficiently demonstrated either claim. With regards to the claim of “moderate levels of responsiveness”: The study should more clearly define the use of the term “moderate”. In the text, however, this is discussed as being indicative of an issue with responsiveness (e.g. “The moderate level of FCPS responsiveness could be attributed to *inadequate* readiness”, “the changes reported in FCPS in the UK and Australia during the pandemic generally reflect the experiences of changes reported in healthcare service [..] exposed the fragility of the current health systems and their unpreparedness to deal with the turbulence of the unexpected”).

However, from what I can see from Fig. 1B, in the UK only in the case of “Choice (in person or virtual)” did a majority of respondents disagree with the statement “I believe my FCP services met the patient's expectations and needs”. In the case of Australia, in none of the criteria a majority of respondents disagreed with that statement. If only in a minority of cases were issues identified with the level of care received by the patients, why is this indicative of an issue with responsiveness? In what way does it reflect “the fragility of the current health systems and their unpreparedness to deal with the turbulence of the unexpected”? With regards to the claim of “inadequate readiness”: It is unclear how the authors determined that both settings demonstrated levels of inadequate readiness; in particular:

Based on Fig 1A, respondents were asked to express their opinion on readiness in 15 different areas. However the results section only reports answers to five (access to information, effective leadership, stakeholder buy-in, tech support, and infrastructure); why was only this subset considered? Why were areas as important as “Outputs & Results” not included? The criteria used to determine whether there is overall agreement with a particular statement are unclear. Based on my understanding, the survey asked respondents whether they “agreed,” “disagreed,” or were “neutral” regarding readiness in various areas. When reporting overall agreement with readiness, the authors appear to use a majority rule—that is, they consider respondents to be in agreement if more than 50% selected “agree”. However, it seems the authors apply a different standard when interpreting disagreement.

Specifically, they appear to treat disagreement with readiness as equivalent to agreement with a lack of readiness (this in itself should be explicitly stated in the text as an assumption), but without requiring a majority for overall interpretation. For example, in support of the statement that “participants agreed there was a lack of readiness for change in terms of infrastructure,” they cite agreement levels of 42.4% (UK) and 39.0% (Australia)—both below the 50% threshold. By contrast, in other areas such as access to information or effective leadership, the authors emphasize whether a majority agreed, suggesting inconsistency in their criteria for interpreting overall agreement levels. I believe their reasoning is that, in the case of the UK, more respondents selected “disagree” than selected “agree” or “neutral” individually. However, this still means that only a minority (<50%) of participants explicitly disagreed with the statement of readiness. Therefore, I do not think the authors can reasonably claim that respondents “agreed that there was lack of readiness for change in terms of infrastructure” based on this result. In the case of Australia, however, I believe the authors have made a mistake when quoting 39% in disagreement with readiness of infrastructure, based on Fig 1A. Please clarify the discrepancy. Based on the subset of 5 areas considered, it would appear that only in 2 out of 5 cases a majority of respondents thought that their FCPS was not ready for rapidly implemented changes. In what way does this support the statement of “inadequate readiness for rapid changes in services”? To establish whether this claim can be supported, authors should 1) use consistent criteria of majority, and 2) summarise all areas investigated in the survey (especially “results and outcomes”) or motivate why only 5/15 are relevant. Overall, I believe the survey results are more nuanced—and at times contradictory to existing literature—than currently acknowledged. The conclusions should reflect this complexity, not just for the sake of accuracy, but also because it adds to the interest and value of the study.

Response: We thank the reviewer for this detailed insight. The paper has now been revised to apply consistent criteria of agreement for determining the majority, using an arbitrary cut-off of ≥50% to indicate moderate to high agreement and <50% for low agreement. The cut-off value has also been included under the data analysis section (page 7). We acknowledge that the reported 39% disagreement regarding infrastructure readiness was an error, and it has been cancelled. All readiness and responsiveness result items have been presented accordingly. The abstract, discussion (pages 19 and 20 on the manuscript with track changes) and conclusion have also been revised accordingly.

The results have been revised as shown on pages 13 and 14 and as below:

Readiness and responsiveness with changes

“In general, strong agreement was consistently reported for the readiness factors of access to information and leadership, with stakeholder’s buy-in recording no agreement across both the UK and Australia. In the UK, participants reported moderate to high agreement (≥50%) with the following four readiness items, presented in descending order of agreement: access to information (72.7%, n = 48), leadership (65.5%, n = 42), mission and vision (55.2%, n = 32), and output and results (51.7%, n = 30). In contrast, the remaining items showed low levels of agreement (<50%), including: compatibility (48.5%, n = 32), infrastructure (39.4%, n = 26), external partnerships and support (39.3%, n = 22), financial resources (38.1%, n = 16), learning strategy (35.7%, n = 20), supplies and procurement (35.5%, n = 32), availability of emergency response guide (35.5%, n = 22), human resources (37.9%, n = 22), human resources management (20.7%, n = 12), technical support (6.1%, n = 4), and stakeholders’ buy-in (0%). In Australia, moderate to high agreement (≥50%) was also reported for three readiness items, listed in descending order: supplies and procurement (73.7%, n = 28), access to information (63.4%, n = 26), and leadership (53.7%, n = 22). All other items recorded lower levels of agreement (<50%), including availability of emergency response guide (48.7%, n = 19), financial resources (43.2%, n = 16), mission and vision (41.0%, n=16), human resources management (46.2%, n=18), learning strategy (40.5%, n=15), human resources (39.0%, n=16), compatibility (32.4%, n=12), external partnership and support (24.2%, n=8), output and results (19.5%, n=8), infrastructure (14.6%, n=6), technical support (9.5%, n=4), and stakeholders’ buy-in (0%).

Regarding responsiveness, communication, autonomy, equitability and prompt attention (timeliness of care) demonstrated notable levels of agreement across both the UK and Australia. Participants from the UK reported moderate to high agreement (≥50%) with seven responsiveness factors, listed in descending order: communication (81.3%, n = 52), autonomy (65.6%, n = 42), equitability (62.1%, n = 36), prompt attention (timeliness of care) (59.4%, n = 38), convenience (56.7%, n = 34), choice (selection of FCPS) (53.1%, n = 34), access to social support (51.7%, n = 30), and respect (50.6%, n = 56). Lower levels of agreement (<50%) were reported for other factors, including prompt attention (investigations) (38.7%, n = 24), choice (in-person vs telehealth) (31.3%, n = 20), and confidentiality (5.9%, n = 4). In contrast, Australian respondents reported moderate to high agreement (≥50%) with only four responsiveness items: communication (90%, n = 36), prompt attention (timeliness of care) (66.7%, n = 28), autonomy (60%, n = 24), and equitability (60.5%, n = 23). The remaining items recorded lower levels of agreement (<50%), including prompt attention (investigations) (42.7%, n = 17), convenience (30%, n = 12), confidentiality (23.7%, n=9), access to social support (23.7%, n=9), choice (selection of FCPS) (18.8%, n=6), choice (in-person vs telehealth) (16.7% (n=5)), respect (7.5%, n=3)”

Minor comments:

- Suggest using “reported changes” in the title

Response: “Reported” has been added to the title

- Fig 1 axes should be number of respondents, not frequency.

Response: Frequency replaced with number of respondents

- Throughout the manuscript, clarify that when analysing responses as a function of country, this is really a function of country + type of service (ED and PC) because of the degeneracy with which the survey was constructed. Maybe this combination could be redefined as “setting”.

Response: The abstract, methods (under data analysis), results and discussion sections have clearly stated that the analysis of survey responses and results presented by country actually reflect a function of both country and their FCPS context (UK [PC] and Australia [ED]).

Reviewer #3: Thanks for your effort to cover the comments but in most cases, the answers and edits made did not cover the reviewer's comments.

Response: Thank you for your review. We note that Reviewer 1 acknowledged that the points raised have been addressed, stating: “The authors have comprehensively addressed the points raised in my first review. I believe currently there is only one major outstanding issue that should be addressed by the authors. I also include a few minor comments.” In the current revision, we have comprehensively addressed the single outstanding issue from Reviewer 1. Therefore, we are unclear about what additional changes Reviewer 3 is requesting, but we would be happy to consider them if we receive further information.

Reviewer #4: Authors carefully responded to Reviewer 1's remarks by revising the title, abstract, introduction, and discussion for clearer definition of the scope of the study on first, perceived, and self-reported changes early in COVID-19 in First Contact Physiotherapy Services (FCPS). They acknowledged limitations such as recall bias, ambiguity in definitions, and sample representativeness. Technical terms such as "adaptations", "timing" were also clarified to prevent misinterpretation. Findings and tables were rewritten to ensure accuracy and unsubstantiated claims were removed and made more specific. Structural and grammatical suggestions for introduction and discussion were applied. Editorial requests were accommodated, such as ethics clarification, participant consent information, data availability via a DOI, and format changes according to PLOS ONE guidelines. Overall, the authors thoroughly revised with substantive changes that sufficiently resolved the reviewer's and editor's concerns and made the manuscript more transparent, consistent, and scientifically valid. Revisions reflect a strong effort to align the content with journal standards and reviewer expectations.

Response: Thank you for taking the time to review our paper and for your generous comments.

Reviewer #5: The definitions of "rapidly implemented changes" and the vagueness of the survey questions could cause respondents to perceive them differently. Concerns around recall bias are further raised by the use of self-reported data. The findings are not well represented in the conclusion, and the authors ought to make a clearer contrast between observed changes in practice and official policy changes. The approach, a cross-sectional survey using physiotherapists' self-reported data, is suitable for documenting perceived changes. However, issues with recall bias and the imprecise definition of "first alterations" could compromise the validity of the results. Overall, even though the goal is pertinent, the method's drawbacks must be carefully considered. Resolving these problems could improve the validity and significance of the study.

Response: Thanks for reviewing our paper. While all these points were addressed in the previous revision (see our earlier responses to reviewers), a summary of those revisions is provided below:

The paper was revised all through to clearly state that the findings are based on providers perceived changes, and the title has also been updated accordingly.

The limitations section clearly highlighted and discussed the recall bias (paged 21 and 22).

In addition, the conclusion has been revised to more accurately reflect the study’s findings (reported changes).

Reviewer #6: Thank you for the invitation to review the manuscript titled “Weathering the storm of COVID-19 pandemic: A cross-sectional survey of changes in first contact physiotherapy services in the UK and Australia” The manuscript is concise and scientifically written using recent literature

I have the following comments to offer in order to improve the quality of the study for use especially during pandemic period to enhance healthcare services in the two regions and globally.

1. Authors should give adequate explanation to changes in service delivery.

Response: Thank you for reviewing our paper. Changes in service delivery is now described in the introduction section (page 4) as below:

“For this study, “changes” refers to the transformations in the delivery of healthcare by FCPS to patients with MSK conditions during the COVID-19 pandemic [33]”.

2. Conclusion in the abstract should highlight the effect on changes in service delivery in relation to future pandemic.

Response: Thank you for your suggestion the conclusion subsection in the abstract has highlighted the effect of changes in the context of managing future pandemic.

3. What are the initial changes noticed in UK and Australia during the pandemic?

Response: The manuscript already highlighted changes in the UK and Australia, and the results presented initial changes in FCPS

4. How did the authors ensure the participants were able to recall changes in their practice during the pandemic, was any instrument used to ascertain cognitive function?

Response: No cognitive function tests were required, as participants were practicing physiotherapists without cognitive impairments. Additionally, recall bias has been thoroughly addressed in the paper, including through survey piloting and discussion of limitations.

5. What is the educational level of the participants?

Response: Many thanks for this comment. Educational level was presented in Table 2 under participant characteristics.

6. The sample size is a bit low to make conclusion based on the findings.

Response: This is an exploratory study which used a non-probability sampling method, reporting perceived changes, hence the sampling is suitable for this type of research [35]. However, sample size has been acknowledged as one of the study limitations on page 22

7. Explain the experiences and qualifications of the 12 participants drawn for the pilot test of the study instrument.

Response: Thank you for the suggestion. The experiences and qualifications of the 12 participants involved in the pilot testing of the study instrument have been included under the section describing the survey instrument (page 5) as below:

“The 12 participants hold master’s degrees, with 75% (n = 9) having more than four years of experience in FCPS, and 25% (n = 3) having three years of experience”

8. In Table 1, the first column should have a heading like item or domain for clarity

Response:

---

## [Decision Letter · Decision Letter 2]

22 Oct 2025

Dear Dr. Adeniji,

Thank you for submitting your manuscript to PLOS ONE. After careful consideration, we feel that it has merit but does not fully meet PLOS ONE’s publication criteria as it currently stands. Therefore, we invite you to submit a revised version of the manuscript that addresses the points raised during the review process.

We look forward to receiving your revised manuscript.

Kind regards,

Aamir Ijaz, MD, FCPS, FRCP, MCPS-HPE

Academic Editor

PLOS ONE

Journal Requirements:

Reviewers' comments:

Reviewer's Responses to Questions

**Comments to the Author**

Reviewer #1: All comments have been addressed

Reviewer #5: All comments have been addressed

2. Is the manuscript technically sound, and do the data support the conclusions?

Reviewer #1: Yes

Reviewer #5: Yes

3. Has the statistical analysis been performed appropriately and rigorously?

Reviewer #1: Yes

Reviewer #5: Yes

4. Have the authors made all data underlying the findings in their manuscript fully available?

Reviewer #1: Yes

Reviewer #5: Yes

5. Is the manuscript presented in an intelligible fashion and written in standard English?

Reviewer #1: Yes

Reviewer #5: Yes

Reviewer #1: (No Response)

Reviewer #5: Clear scope and title updated to “reported changes.”

Abstract now reflects findings accurately and avoids overclaims.

Methods are appropriate and well justified.

Readiness/responsiveness criteria now consistently use ≥50% cut-off as reviewers requested.

All reviewer concerns appear carefully addressed with detailed responses.

Ethics, consent, data availability, conflicts of interest, and CHERRIES checklist are complete.

Tables and figures are reorganized and clarified (e.g. “frequency” → “number of respondents”).

**Do you want your identity to be public for this peer review?** For information about this choice, including consent withdrawal, please see our Privacy Policy

Reviewer #1: No

Reviewer #5: **Yes:**  Ahmed Ibrahim Al Kharusi

---

## [Author Response · Author response to Decision Letter 3]

2 Nov 2025

Thanks to reviewers and editor for reviewing our revised paper.

Editor: The reviewers had no additional comments and confirmed that all previous suggestions had been fully addressed. The final version of the manuscript has now been resubmitted for your consideration.

Thank you for your time and continued support throughout the review process.

---

## [Decision Letter · Decision Letter 3]

30 Dec 2025

Weathering the storm of COVID-19 pandemic: A cross-sectional survey of reported changes in first contact physiotherapy services in the UK and Australia

PONE-D-24-54679R3

Dear Dr. Adeniji,

We’re pleased to inform you that your manuscript has been judged scientifically suitable for publication and will be formally accepted for publication once it meets all outstanding technical requirements.

Kind regards,

Omnia S. El Seifi, M.D., Ph.D.

Academic Editor

PLOS One

Additional Editor Comments (optional):

Reviewers' comments:

Reviewer's Responses to Questions

**Comments to the Author**

Reviewer #5: All comments have been addressed

2. Is the manuscript technically sound, and do the data support the conclusions?

Reviewer #5: Yes

3. Has the statistical analysis been performed appropriately and rigorously?

Reviewer #5: Yes

4. Have the authors made all data underlying the findings in their manuscript fully available?

Reviewer #5: Yes

5. Is the manuscript presented in an intelligible fashion and written in standard English?

Reviewer #5: Yes

Reviewer #5: Thank you for addressing all the comments and now the manuscript is very informative and no more changes required

**Do you want your identity to be public for this peer review?** For information about this choice, including consent withdrawal, please see our Privacy Policy

Reviewer #5: **Yes:**  AHMED IBRAHIM AL KHARUSI

---

## [Editor Report · Acceptance letter]

PONE-D-24-54679R3

PLOS One

Dear Dr. Adeniji,

I'm pleased to inform you that your manuscript has been deemed suitable for publication in PLOS One. Congratulations! Your manuscript is now being handed over to our production team.

Kind regards,

on behalf of

Professor Omnia S. El Seifi

Academic Editor

PLOS One